# TARGET-ORIENTED SINGLE DOMAIN GENERALIZATION

## ABSTRACT

Deep models trained on a single source domain often fail catastrophically under distribution shifts, a critical challenge in Single Domain Generalization (SDG). While existing methods focus on augmenting source data or learning invariant features, they neglect a readily available resource: textual descriptions of the target deployment environment. We propose Target-Oriented Single Domain Generalization (TO-SDG), a novel problem setup that leverages the textual description of the target domain, without requiring any target data, to guide model generalization. To address TO-SDG, we introduce **S**pectral **TAR**get Alignment (STAR), a lightweight module that injects target semantics into source features by exploiting visual-language models (VLMs) such as CLIP. STAR uses a target-anchored subspace derived from the text embedding of the target description to re-center image features toward the deployment domain, then utilizes spectral projection to retain directions aligned with target cues while discarding source-specific noise. Moreover, we use a vision-language distillation to align backbone features with VLM's semantic geometry. STAR further employs feature-space Mixup to ensure smooth transitions between source and target-oriented representations. Experiments across various image classification and object detection benchmarks demonstrate STAR's superiority. This work establishes that minimal textual metadata, which is a practical and often overlooked resource, significantly enhances generalization under severe data constraints, opening new avenues for deploying robust models in target environments with unseen data.

## 1 INTRODUCTION

Deep models trained on data drawn from a single source domain often suffer performance drops when deployed in novel environments that violate their training-distribution assumptions (Volpi et al., 2018; Qiao et al., 2020). In computer vision, for example, classifiers tuned to well-lit daytime photos can misfire on low-light, foggy, or sensor-noisy images that lie just outside their original support. This vulnerability is especially acute in Single-Domain Generalization (SDG), where all training samples originate from one domain and no auxiliary target data are available. Lacking the diversity that multi-domain corpora provide, SDG methods must combat source-specific biases with only limited inductive cues, a challenge that has so far restricted progress relative to classical domain-generalization settings (Gulrajani & Lopez-Paz, 2020).

Rigid SDG protocol overlooks a readily available and inexpensive signal, high-level knowledge of the deployment domain. Even when target images are inaccessible, e.g., for privacy, security, or data-collection cost reasons, practitioners can usually articulate the target domain in words ("radiographs of feline thoraxes", "night-time driving in snow", etc.). Such descriptions encode the semantic factors that drive domain shift but are ignored by traditional SDG pipelines.

Meanwhile, large Vision–Language Models (VLMs) such as CLIP (Radford et al., 2021), ALIGN (Jia et al., 2021), and BLIP (Li et al., 2022) embed both images and natural-language phrases into a shared latent space. Prompt learning (Zhou et al., 2022b), conditional prompting (Zhou et al., 2022a), and adapter layers (Gao et al., 2024; Zhang et al., 2022) show that these embeddings are readily steerable toward new tasks with minimal supervision. While a growing body of CLIP-based work has explored robustness and cross-domain generalization, many methods that explicitly target robustness in domain

generalization settings still rely on target images—, for prompt ensembling, adapter calibration, or test-time refinement, and thus fall outside the strict SDG scenario we consider.

We introduce the new problem of Target-Oriented Single-Domain Generalization (TO-SDG), a protocol that augments the classic SDG setting with one additional asset: a textual specification describing the target environment. TO-SDG explores whether such lightweight metadata can compensate for the absence of target images and, if so, how to inject it into the training loop. To answer this question in image domains, we present Spectral Target Alignment (STAR), a method that infuses target semantics into backbone image features based on VLMs such as CLIP. STAR addresses the core challenge of TO-SGD by integrating target semantics through a unified spectral target orientation process: the natural-language description of the target domain is first encoded into a compact vector via CLIP's text encoder, establishing a semantic anchor in the vision-language space. This vector guides a two-fold transformation of source features, recentering their distribution to align with the target's centroid (mitigating source-specific bias) and projecting them onto a low-rank subspace spanned by the directions most aligned with the target embedding, effectively filtering out noisy, domain-specific variations while preserving discriminative structure. To further harmonize the backbone's geometry with CLIP's semantically rich embedding space, a lightweight distillation loss regresses features toward their CLIP image embedding counterparts, transferring cross-modal invariances without updating the frozen VLM. Finally, feature-space Mixup interpolates between original and target-oriented representations, enforcing smooth decision boundaries that bridge source and target manifolds. The main contributions of the paper are three-fold:

- We introduce the new problem setting of Target-Oriented Single Domain Generalization (TO-SDG), which extends standard SDG by incorporating textual descriptions of the target domain, an accessible and practical source of prior knowledge in real-world deployments.

- We propose Spectral Target Alignment (STAR), a lightweight framework for multiple tasks that injects target semantics via spectral target orientation, vision-language distillation, and feature-space mixup, enabling effective text-guided generalization in TO-SDG.

- Experiments on diverse image classification and object detection benchmarks show that STAR reliably outperforms existing SDG methods, confirming that incorporating freely available natural language descriptions based on various VLMs can significantly improve generalization performance.

## 2 RELATED WORKS

### 2.1 SINGLE DOMAIN GENERALIZATION

Single Domain Generalization (SDG) addresses the formidable challenge of training models that generalize to unseen domains using supervision from only a single source domain, in the absence of any information about target domain distributions. In contrast to conventional domain generalization paradigms that exploit multiple source domains to enhance model robustness, SDG imposes stricter constraints, requiring generalization solely from intra-domain cues. As a result, SDG presents significantly greater difficulty than its multi-source counterpart.

**Classification** Existing approaches to SDG in image classification task primarily focus on enhancing the diversity and expressiveness of the source domain through data augmentation, which can be broadly categorized into three methodological streams. The first stream comprises conventional augmentation techniques aimed at increasing within-domain variability to promote out-of-domain robustness. Techniques such as Improved Regularization via Data Augmentation (DeVries & Taylor, 2017), AugMix (Hendrycks et al., 2019), and AutoAugment (Cubuk et al., 2020a) exemplify this line of work by introducing stochastic or learned perturbations to source samples. Geometric augmentation strategies have been further advanced by Lian et al. (Lian et al., 2021), while ACVC (Cugu et al., 2022) constructs pseudo-domains by applying structured corruptions and aligning attention maps across clean and perturbed images. The second stream adopts adversarial augmentation techniques that generate challenging variants of training samples either in the input or feature space. Early efforts by Volpi et al. (Volpi et al., 2018) and Zhao et al. (Zhao et al., 2020) explore perturbations in the pixel space to induce hard domain shifts. More recent developments, such as Adversarial AutoAugment (Zhang et al., 2019) and the work of Zhang et al. (Zhang et al., 2023a), explore learned augmentation

policies and perturbation of feature statistics, respectively. While these approaches introduce greater diversity, sustaining semantically coherent yet domain-divergent augmentations remains an open challenge. The third stream leverages generative modeling to synthesize novel data distributions. Approaches such as those proposed by Qiao et al. (Qiao et al., 2020), Wang et al. (Wang et al., 2021), and Li et al. (Li et al., 2021) employ GANs and VAEs to generate stylized variants of source data. AdvST(Zheng et al., 2024) treats data augmentations as learnable semantic transformations and uses them adversarially to generate diverse source samples.

**Object Detection** Traditional object detectors such as Faster R-CNN (Ren et al., 2015) form the backbone for most domain generalization studies, but they exhibit sharp performance degradation under domain shifts due to their reliance on source-specific statistics. Early efforts to improve SDG for object detection focused on architectural modifications to enhance representation invariance. IterNorm (Huang et al., 2019) introduces a whitening-based normalization layer to reduce domain-specific covariance, while IBN-Net (Pan et al., 2018) and Switchable Whitening (SW) (Pan et al., 2019) leverage a combination of instance and batch normalization to decouple style and content features. These approaches demonstrated moderate gains but remained limited by their inability to model semantic shifts effectively. More recent work such as ISW (Choi et al., 2021) employs instance style whitening during training to improve robustness, whereas S-DGOD (Wu & Deng, 2022) proposes a specialized architecture tailored to single-source domain generalization by decoupling domain-variant and domain-invariant features. Notably, CLIP-Gap (Vidit et al., 2023) incorporates pretrained VLMs to bridge the semantic gap between source and target domains, demonstrating the potential of aligning high-level semantic features with textual priors in detection tasks. These methods highlight the progression from normalization-based heuristics to semantically guided feature adaptation, marking a significant shift in SDG strategies for object detection.

## 2.2 EXPLOITATION OF VISION-LANGUAGE MODELS

The advent of foundational VLMs such as CLIP (Radford et al., 2021) has redefined the landscape of representation learning, offering pre-trained multimodal encoders that exhibit strong transfer capabilities across a variety of downstream tasks. Notably, the text–image alignment learned by CLIP provides a rich semantic embedding space that remains stable across moderate domain shifts, making it a natural candidate for domain generalization (DG). Recent efforts have explored integrating CLIP within DG pipelines to harness its robustness. CoCoOp (Zhou et al., 2022a) introduces a prompt-tuning mechanism conditioned on visual features, showing that dynamically adapting textual prompts can significantly improve zero-shot generalization. CLIP-Adapter (Gao et al., 2024) and Tip-Adapter (Zhang et al., 2022) extend this line by proposing lightweight residual pathways that adapt CLIP to novel distributions while retaining its pretrained alignment. In parallel, Fort et al. (Fort et al., 2021) demonstrate that CLIP embeddings serve as effective detectors of distributional shift, outperforming dedicated uncertainty estimation techniques in out-of-distribution (OOD) detection. VLMs have been employed to synthesize or hallucinate domain-variant supervision. UniDG (Zhang et al., 2023b) leverages CLIP to generate text-guided augmentations that simulate domain shifts, while recent retrieval-based methods (Shu et al., 2023) use textual descriptions to anchor unseen domains in the shared embedding space. Collectively, these works suggest that vision–language models not only encode invariant semantics but also offer actionable priors that downstream models can exploit to remain robust under distribution shift. While CLIP remains the dominant vision–language model in domain generalization research, owing to its open accessibility and robust zero-shot transfer, emerging work has begun to explore alternatives. Addepalli et al. (Addepalli et al., 2024) propose VL2V-SD and VL2V-ADiP, incorporating BLIP-2 (Li et al., 2022) into self-distillation pipelines to improve out-of-distribution generalization in image classification. Despite their promise, such models remain underutilized in DG frameworks, largely due to limited public availability or incompatibilities with standard prompt-based architectures, which are tightly coupled with CLIP's dual encoder design.

## 2.3 ZERO-SHOT DOMAIN ADAPTATION

Zero-shot domain adaptation (ZSDA) aims to transfer from a labeled source to an unlabeled target domain without task-relevant target labels, often exploiting auxiliary target-side information. Our Target-Oriented Single Domain Generalization (TO-SDG) differs from ZSDA formulation in as-

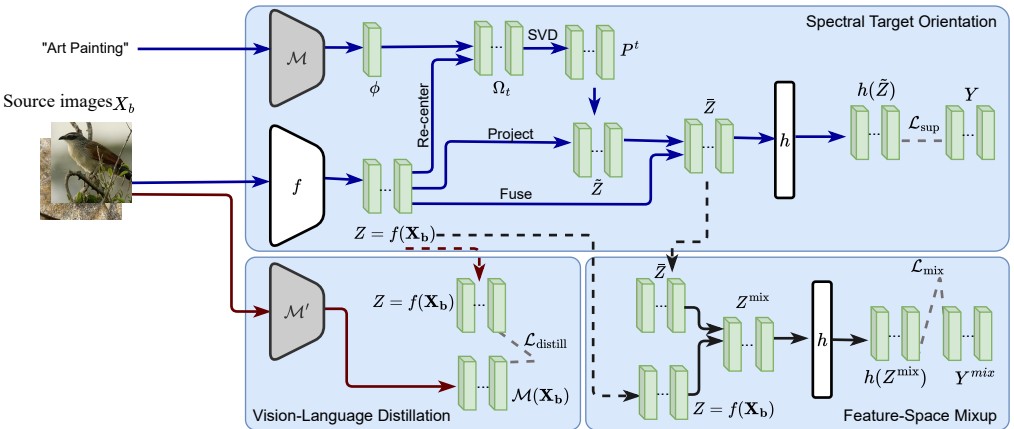

Figure 1: An illustration of the STAR framework with three components: Spectral Target Orientation (STO), Vision-Language Distillation (VLD), and Feature-Space Mixup (FSM). Given source images and a target text description as input, image features are extracted via a backbone network $f$, while the target description is mapped to a domain embedding $\phi^t$ using a frozen VLM text encoder $\mathcal{M}$. In STO, backbone features are re-centered around $\phi^t$ and projected onto a target-aligned subspace produced via SVD decomposition. The resulting features are fused with the original representations for supervised loss $\mathcal{L}_{\text{sup}}$ calculation. In VLD, backbone features are aligned with the image embeddings produced with a frozen VLM image encoder $\mathcal{M}'$, providing semantic guidance through the distillation loss $\mathcal{L}_{\text{distill}}$. In FSM, feature space mixup is conducted to encourage smooth semantic transition across domains with the mixup loss $\mathcal{L}_{\text{mix}}$.

sumptions. We assume a single labeled source domain and *no* target-domain images—task-relevant, task-irrelevant, or synthesized—during training; the only target-side signal is a high-level textual description of the deployment domain. Early work such as Zero-Shot Deep Domain Adaptation (Peng et al., 2018) synthesizes target-style features via task-irrelevant dual-domain pairs. More recent methods inject language guidance or prompts into this pipeline: Unified Language-driven ZSDA (Yang et al., 2024) and PØDA (Fahes et al., 2023) use text prompts and CLIP-style encoders to simulate or rectify target-like features; For object detection, zero-shot day–night DA (Du et al., 2024) and UPRE (Zhang et al., 2025) similarly boost detectors by constructing target-aware representations (e.g., via Retinex-inspired decompositions or unified prompts) despite the lack of labeled target data.

## 3 METHOD

### 3.1 PROBLEM SETUP

We propose Target-Oriented Single Domain Generalization (TO-SDG), a novel setting where the goal is to leverage source domain data alongside a high-level specification of the target domain. Formally, we are given a labeled dataset $\mathcal{D}^s = \{(\mathbf{x}_i, \mathbf{y}_i)\}_{i=1}^{N}$ from a single source domain, where $\mathbf{x}_i \in \mathcal{X}$ and $\mathbf{y}_i \in \mathcal{Y}$. In addition, a textual description of the unseen target domain is provided, denoted by $T$. The objective is to learn a predictive model $(h \circ f) : \mathcal{X} \to \mathcal{Y}$ that generalizes to the specified target domain, whose data remain unavailable during training. The model is composed of a feature extractor $f_\theta : \mathcal{X} \to \mathcal{Z}$ parameterized by $\theta$, and a predictor $h_\psi : \mathcal{Z} \to \mathcal{Y}$ parameterized by $\psi$. The key challenge lies in utilizing the auxiliary target description $T$ to guide the learning process and promote generalization under domain shift.

### 3.2 PROPOSED METHOD

To tackle TO-SDG, we propose STAR, a framework that injects target-specific semantic priors into the backbone feature space via a vision–language model such as CLIP and refines those representations through spectral editing and regularization. STAR consists of three key components. First, Spectral Target Orientation (STO) uses a frozen CLIP text encoder to extract a target domain embedding from the textual description. This embedding anchors the backbone feature orientation through

first-order translation and spectral projection onto a low-rank, target-aligned subspace, promoting generalizability towards the target domain. Second, Vision-Language Distillation (VLD) transfers semantic structure from CLIP image embeddings into the backbone features, aligning the representation geometry with that of a semantically rich, pretrained VLM. Third, Feature-Space Mixup (FSM) interpolates between original and target-oriented features to smooth transitions across domains. The overall framework is illustrated in Figure 1.

### 3.2.1 SPECTRAL TARGET ORIENTATION

Robust generalisation from a single observed domain demands that the backbone feature extractor $f_\theta$ acquires a representation that is anchored to the semantics of the target deployment environment while simultaneously remaining faithful to the discriminative structure of the source data. We operationalize this intuition through Spectral Target Orientation (STO) a two-stage procedure that (i) re-centres batch statistics around a language-derived target anchor and (ii) projects the resulting features onto a low-rank subspace whose directions are maximally aligned with that anchor.

**Target Anchor Guided Translation** The procedure begins with a free-form textual description $T$ of the target environment. A frozen text encoder $\mathcal{M}$ from a vision-language model, specifically CLIP, maps this description into the multimodal latent space, producing a target embedding

$$\phi^t = \mathcal{M}(T), \qquad \phi^t \in \mathbb{R}^{1 \times d}, \tag{1}$$

where $d$ denotes the embedding dimension shared with the image encoder. The vector $\phi^t$ serves as an external semantic prior for the target domain: it encapsulates those high-level attributes that are expected to shift between source and target, and it does so without requiring any target images.

Given a mini-batch $X_b = \{\mathbf{x}_i\}_{i=1}^{n_b}$ of source samples, the backbone produces the feature matrix $Z = f_\theta(X_b)$. Let $\mathbf{z}_i$ denote the $i$-th row of $Z$ and $\mu^z = \frac{1}{n_b} \sum_{i=1}^{n_b} \mathbf{z}_i$ be the batch mean. Classical style-transfer (Huang & Belongie, 2017) shows that first-order statistics largely account for appearance discrepancies; we therefore translate the batch so its centre of mass coincides with the target anchor:

$$\Omega_i^t = (\mathbf{z}_i - \mu^z) + \phi^t. \tag{2}$$

where $\Omega_i^t$ is the $i$-th row of the translated feature matrix $\Omega^t$. This affine shift removes source-specific bias while injecting target semantics. Crucially, because $\phi^t$ originates from the same CLIP space as the image embeddings, the shift operates in a modality-aligned coordinate system, avoiding the semantic drift that often plagues purely heuristic normalization schemes.

**Spectral Target-Oriented Projection** Although the mean shift corrects first-order statistics, the features may still encode nuisances correlated with the source domain. To disentangle semantically meaningful factors from such noise, we perform singular value decomposition (SVD) on the recentered features:

$$\Omega^t = U^t \, \Sigma^t \, (V^t)^\top, \tag{3}$$

where $U^t \in \mathbb{R}^{n_b \times n_b}$ and $V^t \in \mathbb{R}^{d \times d}$ are orthonormal and $\Sigma^t$ is a diagonal-like matrix with non-negative singular values. Each column of $V^t$ provides an orthogonal direction in feature space, ranked by how strongly it varies after re-centring around $\phi^t$. Retaining only the top $k = \eta \cdot d$ directions, where $0 < \eta < 1$, we construct the truncated basis $V_k^t \in \mathbb{R}^{d \times k}$ that spans a target-aligned subspace, discarding low-variance noise while preserving transferable semantics. We map the backbone features into this subspace using the following projection matrix:

$$P^t = V_k^t (V_k^t)^\top, \tag{4}$$

thereby isolating a low-rank subspace that captures target-aligned variance while suppressing low-energy directions dominated by source-domain noise. Projecting the original feature batch, renormalizing and rescaling each sample gives target oriented features

$$\tilde{Z} = \frac{ZP^t}{\|ZP^t\|_2} \, \|Z\|_2, \tag{5}$$

where $\|\cdot\|_2$ denotes the $\ell_2$ norm. The final rescaling preserves the norm of each batch, preventing the downstream predictor from the impact of varied amplitudes.

Table 1: Classification accuracy (%) and standard deviation on the PACS dataset using "Photo" as the source domain. Each column corresponds to a different target domain.

| Method | Art | Cartoon | Sketch | Avg. |
|---|---|---|---|---|
| MixUp (Zhang et al., 2018) | 52.8 | 17.0 | 23.2 | 31.0 |
| CutOut (DeVries & Taylor, 2017) | 59.8 | 21.6 | 28.8 | 36.7 |
| ADA (Volpi et al., 2018) | 58.0 | 25.3 | 30.1 | 37.8 |
| ME-ADA (Zhao et al., 2020) | 60.7 | 28.5 | 29.6 | 39.6 |
| AugMix (Hendrycks et al., 2019) | 63.9 | 27.7 | 30.9 | 40.8 |
| RandAug (Cubuk et al., 2020b) | 67.8 | 28.9 | 37.0 | 44.6 |
| ACVC (Cugu et al., 2022) | 67.8 | 30.3 | 46.4 | 48.2 |
| L2D (Wang et al., 2021) | 67.6 | 42.6 | 47.1 | 52.5 |
| MCL (Chen et al., 2023) | - | - | - | 59.6 |
| Prompt-Driven (Li et al., 2024) | - | - | - | 60.0 |
| STAR (Ours) | $\mathbf{75.3}_{(0.4)}$ | $\mathbf{52.5}_{(0.3)}$ | $\mathbf{53.1}_{(0.4)}$ | $\mathbf{60.3}$ |

Table 2: Classification accuracy and standard deviation(%) on the DomainNet dataset using "Real" as the source domain. Each column corresponds to a different target domain.

| Method | Painting | Infograph | Clipart | Sketch | Quickdraw | Avg. |
|---|---|---|---|---|---|---|
| MixUp (Zhang et al., 2018) | 38.6 | 13.9 | 38.0 | 26.0 | 3.7 | 24.0 |
| CutOut (DeVries & Taylor, 2017) | 38.3 | 13.7 | 38.4 | 26.2 | 3.7 | 24.1 |
| CutMix (Yun et al., 2019) | 38.3 | 13.5 | 38.7 | 26.9 | 3.6 | 24.2 |
| ADA (Volpi et al., 2018) | 38.2 | 13.8 | 40.2 | 24.8 | 4.3 | 24.3 |
| ME-ADA (Zhao et al., 2020) | 39.0 | 14.0 | 41.0 | 25.3 | 4.3 | 24.7 |
| RandAug (Cubuk et al., 2020b) | 41.3 | 13.6 | 41.1 | 30.4 | 5.3 | 26.3 |
| AugMix(Hendrycks et al., 2019) | 40.8 | 13.9 | 41.7 | 29.8 | 6.3 | 26.5 |
| ACVC (Cugu et al., 2022) | 41.3 | 12.9 | 42.8 | 30.9 | 6.6 | 26.9 |
| AdvST (Zheng et al., 2024) | $42.3_{(0.1)}$ | $14.8_{(0.2)}$ | $43.5_{(0.4)}$ | $30.8_{(0.3)}$ | $5.9_{(0.2)}$ | 27.1 |
| STAR (Ours) | $\mathbf{45.9}_{(0.1)}$ | $\mathbf{17.1}_{(0.3)}$ | $\mathbf{45.7}_{(0.6)}$ | $\mathbf{35.2}_{(0.5)}$ | $\mathbf{9.2}_{(0.3)}$ | $\mathbf{30.0}$ |

Generalization can further benefit from stochastic feature-space augmentation. Specifically, we apply standard weak image transformation on $X_b$ to obtain $\bar{X}_b$, which is then processed identically as $X_b$ using Eq 2–Eq 5 to produce $\bar{Z}$. Averaging the two views,

$$\tilde{Z}_{\text{avg}} = (\tilde{Z} + \bar{Z})/2, \tag{6}$$

smooths high-frequency artefacts introduced by augmentation and encourages the model to remain invariant along the geodesic connecting them in representation space.

Finally, we blend the target-oriented features with the original representations,

$$\hat{Z} = (1 - \alpha)\, Z + \alpha\, \tilde{Z}_{\text{avg}}, \tag{7}$$

with $\alpha \in [0, 1]$ modulating the strength of orientation. For convenience, the entire STO transformation via Eq 2–Eq 7 on a single sample $\mathbf{x}$ within its batch can be written compactly as

$$\hat{\mathbf{z}} = g\big(f_\theta(\mathbf{x}), \phi^t\big), \tag{8}$$

where $g(\cdot, \cdot)$ denotes the STO operator. This operation modulates the feature representation by integrating semantic guidance from the target domain embedding $\phi^t$, and the target-oriented features are then used to minimize the supervised empirical loss:

$$\mathcal{L}_{\text{sup}}(\theta, \psi) = \mathbb{E}_{(\mathbf{x},\mathbf{y}) \in \mathcal{D}^s}\Big[ \ell_{\text{sup}}\big(h_\psi(g(f_\theta(\mathbf{x}), \phi^t)), \mathbf{y}\big) \Big] \tag{9}$$

where $\ell_{\text{sup}}$ denotes the standard supervised loss—e.g., a cross-entropy loss for the classification task.

STO imposes a structured combination of operation steps on the source data features: a first-order translation guided by the target embedding $\phi^t$, a spectral projection step that selectively eliminates source-specific variance, and a controlled blending of oriented and original features. Together with supervised learning in the source domain, these operation steps produce representations that reside on a submanifold optimized for maintaining discriminative performance on the source domain while effectively aligning with the semantics of the unseen target domain. Consequently, STO provides a principled mechanism for achieving robust generalization in the challenging TO-SDG setting.

### 3.2.2 VISION–LANGUAGE DISTILLATION

While STAR's spectral target orientation steers features toward the direction of the target domain, it does not by itself guarantee that the resulting representation retains the semantic structure captured by vision-language models–an essential property for meaningful and effective spectral target orientation. We therefore introduce an auxiliary vision-language distillation term that transfers high-level semantics from the fixed image encoder $\mathcal{M}'$ of the adopted vision-language model, CLIP. For every source image $\mathbf{x}$, we regress the backbone feature output $f_\theta(\mathbf{x})$ onto its CLIP image embedding $\mathcal{M}'(\mathbf{x})$ via a squared distillation loss

$$\mathcal{L}_{\text{distill}}(\theta) = \mathbb{E}_{\mathbf{x} \in \mathcal{D}^s} \| f_\theta(\mathbf{x}) - \mathcal{M}'(\mathbf{x}) \|_2^2. \tag{10}$$

This coupling encourages the backbone representation to inherit CLIP's global semantic geometry, such as intra-class attributes and inter-class structures, without updating CLIP or accessing any target images. Empirically, the distillation term synergizes with STAR's domain-aware editing, yielding representations that are simultaneously target-oriented and semantically well calibrated, thereby boosting generalizability to the unseen target domain.

### 3.2.3 FEATURE-SPACE MIXUP

To enhance generalizability over the prediction model and promote locally linear behaviour as features traverse from source-biased to target-aligned regions of the representation manifold, we adopt a feature–space Mixup strategy that operates after STAR's spectral target orientation. Concretely, for each batch we apply a random permutation to the target-oriented samples to obtain a re-indexed batch $\{\hat{\mathbf{z}}_i'\}_{i=1}^{n_b}$ along with their corresponding one-hot labels $\{\mathbf{y}_i'\}_{i=1}^{n_b}$, then form convex combinations

$$\mathbf{z}_i^{\text{mix}} = \beta \mathbf{z}_i + (1 - \beta) \hat{\mathbf{z}}_i', \qquad \mathbf{y}_i^{\text{mix}} = \beta \mathbf{y}_i + (1 - \beta) \mathbf{y}_i', \tag{11}$$

where $\beta$, sampled from a Beta distribution, controls the interpolation strength. Mixing in latent space, rather than pixels, couples the original source representation $\mathbf{z}_i = f_\theta(\mathbf{x}_i)$ with the target-oriented representation $\hat{\mathbf{z}}_i' = g(f_\theta(\mathbf{x}_i'), \phi^t)$. The resulted mixup set $\mathcal{D}^{\text{mix}} = \{(\mathbf{z}_i^{\text{mix}}, \mathbf{y}_i^{\text{mix}})\}_{i=1}^{n_b}$ feeds a standard supervised objective

$$\mathcal{L}_{\text{mix}}(\theta, \psi) = \mathbb{E}_{(\mathbf{z}^{\text{mix}}, \mathbf{y}^{\text{mix}}) \in \mathcal{D}^{\text{mix}}} \left[ \ell_{\text{sup}} \left( h_\psi(\mathbf{z}^{\text{mix}}), \mathbf{y}^{\text{mix}} \right) \right], \tag{12}$$

which synergizes with the primary supervised loss and distillation term to compel the decision surface to vary smoothly along semantic trajectories that interpolate between source and target cues and to foster representations that extrapolate gracefully to the unseen target domain.

### 3.2.4 OVERALL OBJECTIVE

The complete training objective integrates supervised and regularization terms to jointly optimize discriminative performance and semantic alignment:

$$\mathcal{L}_{\text{tr}} = \mathcal{L}_{\text{sup}} + \lambda_{\text{mix}} \mathcal{L}_{\text{mix}} + \lambda_{\text{distill}} \mathcal{L}_{\text{distill}}, \tag{13}$$

where $\lambda_{\text{mix}}$ and $\lambda_{\text{distill}}$ are trade-off hyperparameters that control the influence of the feature-space Mixup supervision and vision–language distillation losses, respectively. This composite objective enforces that the model learns features that are both robust to distributional shifts and semantically consistent with the target domain.

## 4 EXPERIMENTS

### 4.1 EXPERIMENTAL SETUP

**Datasets** We evaluate on three benchmarks with significant domain shift. PACS Li et al. (2017): four stylistic domains (Art, Cartoon, Photo, Sketch) with seven classes; Photo is the source, others are targets. DomainNet Peng et al. (2019): six heterogeneous domains (Real, Infograph, Clipart, Painting, Quickdraw, Sketch) with 345 classes; Real is the source, others are targets. Diverse-Weather Wu & Deng (2022): five urban scene domains (Daytime Clear, Night Clear, Dusk Rainy, Night Rainy, Daytime Foggy); Daytime Clear is the source

Table 3: Mean Average Precision (mAP) results on the Diverse-Weather dataset with "Day Clear" as the source domain. Each column represents performance on a different target domain.

| Method | Day Clear | Night Clear | Dusk Rainy | Night Rainy | Day Foggy |
|---|---|---|---|---|---|
| Faster-RCNN Ren et al. (2015) | 48.1 | 34.4 | 26.0 | 12.4 | 32.0 |
| IterNorm Huang et al. (2019) | 43.9 | 29.6 | 22.8 | 12.6 | 28.4 |
| SW Pan et al. (2019) | 50.6 | 33.4 | 26.3 | 13.7 | 30.8 |
| IBN-Net Pan et al. (2018) | 49.7 | 32.1 | 26.1 | 14.3 | 29.6 |
| ISW Choi et al. (2021) | 51.3 | 33.2 | 25.9 | 14.1 | 31.8 |
| S-DGOD Wu & Deng (2022) | 56.1 | 36.6 | 28.2 | 16.6 | 33.5 |
| CLIP-Gap Vidit et al. (2023) | 51.3 | 36.9 | 32.3 | 18.7 | 38.5 |
| Prompt-Driven Li et al. (2024) | 53.6 | 38.5 | 33.7 | 19.2 | 39.1 |
| STAR (Ours) | **58.3** | **39.5** | **35.2** | **21.0** | **40.4** |

Table 4: Per-class Average Precision (AP) results for object detection on the Diverse-Weather dataset, using "Day Clear" as the source domain and "Dusk Rainy" as the target domain.

| Method | bus | bike | car | motor | person | rider | truck | mAP |
|---|---|---|---|---|---|---|---|---|
| Faster-RCNN Ren et al. (2015) | 28.5 | 20.3 | 58.2 | 6.5 | 23.4 | 11.3 | 33.9 | 26.0 |
| IterNorm Huang et al. (2019) | 32.9 | 14.1 | 38.9 | 11.0 | 15.5 | 11.6 | 35.7 | 22.8 |
| SW Pan et al. (2019) | 35.2 | 16.7 | 50.1 | 10.4 | 20.1 | 13.0 | 38.8 | 26.3 |
| IBN-Net Pan et al. (2018) | 37.0 | 14.8 | 50.3 | 11.4 | 17.3 | 13.3 | 38.4 | 26.1 |
| ISW Choi et al. (2021) | 34.7 | 16.0 | 50.0 | 11.1 | 17.8 | 12.6 | 38.8 | 25.9 |
| S-DGOD Wu & Deng (2022) | 37.1 | 19.6 | 50.9 | 13.4 | 19.7 | 16.3 | 40.7 | 28.2 |
| CLIP-Gap Vidit et al. (2023) | 37.8 | 22.8 | 60.7 | 16.8 | 26.8 | 18.7 | 42.4 | 32.3 |
| Prompt-Driven Li et al. (2024) | 39.4 | 25.2 | 60.9 | 20.4 | 29.9 | 16.5 | **43.9** | 33.7 |
| Ours | **41.4** | **26.1** | **62.7** | **21.9** | **30.0** | **20.5** | 43.8 | **35.2** |

**Experimental Details**   Our experiments follow the standard setup used in prior SDG studies for image classification and object detection Zheng et al. (2024); Wu & Deng (2022). Specifically for STAR, we use the pretrained CLIP ViT-B/32 model as the vision–language backbone, and in our ablations we additionally evaluate LLaVA-1.5 (7B) and BLIP ViT-B/16. . For each target domain, the corresponding domain name (e.g., "Art Painting", "Night Rainy", etc.), is used as the textual description $T$. We set the STO blending coefficient $\alpha = 0.9$, the Mixup weight $\lambda_{\mathrm{mix}} = 0.5$, the vision–language alignment weight $\lambda_{\mathrm{distill}} = 0.01$, and the spectral truncation ratio $\eta = 0.5$. Detailed configurations are provided in Supplementary Materials.

## 4.2 COMPARISON RESULTS

We evaluate our method against a broad spectrum of baselines. For image classification, we compare with MixUp Zhang et al. (2018), CutOut DeVries & Taylor (2017), CutMix Yun et al. (2019), AutoAugment Cubuk et al. (2018), RandAugment Cubuk et al. (2020b), and AugMix Hendrycks et al. (2019), ACVC Cugu et al. (2022), ERM Koltchinskii (2011), CCSA Motiian et al. (2017), JiGen Carlucci et al. (2019), ADA Volpi et al. (2018), ME-ADA Zhao et al. (2020), RSDA Volpi & Murino (2019), L2D Wang et al. (2021), PDEN Li et al. (2021), MCL Chen et al. (2023), Prompt-Driven Li et al. (2024), and AdvST Zheng et al. (2024). All classification experiments are conducted using a ResNet-18 backbone. For object detection, we evaluate our framework against established baselines and DG-specific variants. We include Faster R-CNN Ren et al. (2015), IterNorm Huang et al. (2019), Switchable Whitening (SW) Pan et al. (2019), IBN-Net Pan et al. (2018), Instance Style Whitening (ISW) Choi et al. (2021), S-DGOD Wu & Deng (2022) and CLIP-Gap Vidit et al. (2023), Prompt-Driven Li et al. (2024), implemented with a ResNet-101 backbone.

Table 1 presents the classification results on the PACS dataset using ResNet-18 as the backbone. Our proposed method, STAR, consistently outperforms prior approaches across all target domains, achieving the highest average accuracy of 60.3%, surpassing the next-best method PR-C by a notable margin of 3.2%. STAR sets a new state of the art in the "Art" domain with 75.3%, outperforming ACVC; achieves 52.5% on "Cartoon," exceeding L2D by 9.9%; and reaches 53.1% on "Sketch," outperforming L2D by 6.0%. These gains highlight STAR's ability to robustly generalize under severe domain shifts.

Table 2 reports performance on the DomainNet benchmark with ResNet-18 as the backbone. STAR achieves the highest average accuracy of 30.0%, outperforming the strongest prior method, AdvST,

Table 5: Ablation studies classification on different VLM choice. Accuracy and standard deviation(%) comparison on the PACS dataset.

| Target | $-$w BLIP | $-$w LLaVA | - w CLIP  (Ours) |
|---|---|---|---|
| Art | $74.1_{(0.6)}$ | $73.9_{(0.8)}$ | $75.3_{(0.4)}$ |
| Cartoon | $51.3_{(0.4)}$ | $52.0_{(0.7)}$ | $51.5_{(0.3)}$ |
| Sketch | $51.6_{(0.6)}$ | $51.1_{(0.7)}$ | $52.7_{(0.4)}$ |
| Avg. | 59.0 | 59.0 | 59.8 |

Table 6: Ablation studies classification accuracy and standard deviation(%) comparison on the PACS dataset using "Photo" as the source domain. Each row corresponds to a different target domain.

| Target | $-$w/o $\mathcal{L}_{\text{distill}}$ | $-$w/o $\mathcal{L}_{\text{sup}}$ | $-$w/o $\mathcal{L}_{\text{mix}}$ | $-$w/o projection | $-$w bottom $k$ | $-$w/o Aug | STAR (Ours) |
|---|---|---|---|---|---|---|---|
| Art | $69.7_{(0.5)}$ | $49.1_{(0.9)}$ | $70.1_{(0.5)}$ | $65.9_{(0.9)}$ | $65.9_{(0.9)}$ | $73.2_{(0.5)}$ | $\mathbf{75.3}_{(0.4)}$ |
| Cartoon | $48.3_{(0.4)}$ | $38.7_{(0.6)}$ | $49.0_{(0.5)}$ | $43.6_{(0.7)}$ | $49.0_{(0.8)}$ | $49.6_{(0.8)}$ | $\mathbf{52.5}_{(0.3)}$ |
| Sketch | $49.6_{(0.6)}$ | $39.6_{(0.5)}$ | $50.6_{(0.1)}$ | $44.0_{(0.6)}$ | $49.8_{(0.5)}$ | $50.1_{(0.7)}$ | $\mathbf{53.1}_{(0.4)}$ |
| Avg. | 55.8 | 42.4 | 56.5 | 51.1 | 54.9 | 57.6 | $\mathbf{60.3}$ |

across all five target domains. The most pronounced gain is observed in the "Sketch" domain, where STAR improves accuracy by 4.4%, highlighting its robustness under severe domain shifts.

Table 3 reports single-domain generalization performance for object detection on the Diverse-Weather benchmark, measured in mean Average Precision (mAP) across five unseen target domains. STAR outperforms all prior methods by a clear margin across all conditions, including severe low-light and adverse weather scenarios. Compared to the strongest baseline, CLIP-Gap, which leverages vision–language pretraining to mitigate domain shift, STAR delivers consistent and sizable improvements. On "Day Foggy," STAR achieves 40.4 mAP, outperforming CLIP-Gap by 1.9 points; on the challenging "Night Rainy" setting, STAR reaches 21.0 mAP, a 2.3-point gain over CLIP-Gap. Notably, in "Dusk Rainy" and "Night Clear," STAR surpasses CLIP-Gap by 2.9 and 2.6 points respectively, reflecting superior robustness to both illumination variance and atmospheric degradation.

Table 4 reports object detection performance on the Dusk Rainy scene, which presents a substantial domain shift from the source (Daytime Clear) due to the joint challenges of low visibility and rain-induced noise. While performance across most object categories remains competitive with prior methods, STAR demonstrates clear improvements in key dynamic object classes, achieving gains of 3.6% for "bus", 4.8% for "motor", and 3.2% for "person". Additional per-class object detection results are presented in the Appendix.

### 4.3 Ablation Studies

#### 4.3.1 Ablation on Vision–Language Models

We conduct an ablation study to assess the effect of different target text encoders on STAR's performance. Specifically, we compare three categories of pretrained models for deriving the domain embedding $\phi^t$: BLIP Li et al. (2022), a vision–language model optimized for image–text alignment; LLaVA Liu et al. (2023), a multimodal large language model (MLLM) tuned for visual instruction-following; and CLIP, our default choice, which is pretrained with contrastive supervision over image–text pairs at scale. Table 5 summarizes classification accuracy across target domains in PACS. While all variants perform competitively, CLIP consistently yields the highest average accuracy (59.8%), outperforming BLIP and LLaVA by 0.8% on average. Notably, CLIP excels in the Art and Sketch domains, suggesting its embeddings more faithfully preserve visual-domain semantics that are useful for feature alignment. Although LLaVA slightly outperforms other models on the Cartoon domain, it exhibits higher variance and underperforms on other shifts, likely due to its instruction-tuning objective being less aligned with low-level visual representation.

#### 4.3.2 Ablation on Different Components

Table 6 presents a comprehensive ablation study assessing the contribution of each component in the STAR framework on PACS dataset by using Resnet-18 as the backbone network. In particular, we compared the full model STAR with the following variants: (1) "$-$w/o $\mathcal{L}_{\text{distill}}$" which drops the distillation loss; (2) "$-$w/o $\mathcal{L}_{\text{sup}}$" which omits the primary classification objective; (3) "$-$w/o $\mathcal{L}_{\text{mix}}$" which disables feature-space Mixup; (4) "$-$w/o projection" which bypasses the spectral target-oriented,

keeping re-centered features without projecting onto the target-aligned subspace; (5) "−w bottom $k$" which replaces the top-$k$ projection with a projection onto the lowest-variance components, testing whether performance gains arise from dimensionality reduction alone or from semantically meaningful axes; and (6) "−w/o Aug" which removes data augmentation. Removing the classification loss $\mathcal{L}_{\text{sup}}$, which serves as the primary supervised signal, results in a dramatic degradation in performance, with an average accuracy of just 42.4%, confirming its foundational role in grounding the target-conditioned features with task-specific supervision. Excluding the vision–language distillation term $\mathcal{L}_{\text{distill}}$ leads to a 4-point drop in average accuracy (from 59.8% to 55.8%), indicating that semantic guidance from the CLIP teacher effectively regularizes the feature space and enhances cross-domain alignment. Spectral projection is crucial. Removing it drops accuracy to 51.1%, showing the value of subspace filtering. A bottom-$k$ projection performs worse (54.9%) than top-$k$, confirming that principal components capture the key semantic variation Disabling Mixup ($\mathcal{L}_{\text{mix}}$) reduces accuracy to 56.5%, underscoring its role in source–target alignment. Removing image-space augmentation further drops it to 57.6%, showing that even mild perturbations stabilize the target subspace.

## 5 CONCLUSION

In this paper, we introduce Target-Oriented Single Domain Generalization (TO-SDG), a new problem setup that augments the standard SDG problem with a simple natural language text description of the target domain. To address the challenges of TO-SDG, we propose a novel approach, Spectral Target Alignment (STAR), which aligns source features with target deployment-domain semantics via text-guided spectral target orientation, vision-language distillation, and feature-space Mixup augmentation, effectively purging source-specific biases while preserving discriminative structure. Experiments across image classification and object detection benchmarks demonstrate STAR's superiority over prior methods. By harnessing freely available textual metadata, STAR establishes a practical pathway toward robust model deployment in unseen domains.

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

---

**Algorithm 1: Training Algorithm for STAR**

---

**Input**: Source dataset $\mathcal{D}^s$, target text $T$, $\mathcal{M}$, $\mathcal{M}'$ $f_{\theta_0}$ and $h_{\psi_0}$
**Output**: Trained prediction model $f_\theta$, $h_\psi$

Compute target embedding $\phi^t = \mathcal{M}(T)$ using Eq. (1)
**for** Iteration $i = 1$ to $I$ **do**
  **for** Batch $(X_b, Y_b)$ in $\mathcal{D}^s$ **do**
    Compute features $Z = f_\theta(X_b)$, batch mean $\mu^z$, and recenter to obtain $\Omega^t$ using Eq. (2)
    Compute SVD on $\Omega^t$ using Eq. (3)
    Construct projection matrix $P^t$ using Eq. (4)
    Project features $Z$ via $P^t$ using Eq. (5)
    Compute $\bar{Z}$ on augmented version of $X_b$ using Eq. (3), Eq. (4), and Eq.(5)
    Average $\bar{Z}$ and $\tilde{Z}$ using Eq. (6)
    Compute target-conditioned features $\hat{Z}$ using Eq.(7)
    Compute classification loss $\mathcal{L}_{\text{sup}}$ on $\hat{Z}$ using Eq. (9)
    Compute distillation loss $\mathcal{L}_{\text{distill}}$ on $X_b$ using Eq. (10)
    Generate mixup data $\mathcal{D}^{\text{mix}}$ and compute $\mathcal{L}_{\text{mix}}$ on $\mathcal{D}^{\text{mix}}$ using Eq. (11) and Eq. (12)
    Update $\theta$, $\psi$ via gradient descent on $\mathcal{L} = \mathcal{L}_{\text{sup}} + \lambda_{\text{mix}}\mathcal{L}_{\text{mix}} + \lambda_{\text{distill}}\mathcal{L}_{\text{distill}}$
  **end for**
**end for**

---

Table 7: Per-class Average Precision (AP) results for object detection on the Diverse-Weather dataset, using "Day Clear" as the source domain and "Night Clear" as the target domain.

| Method | bus | bike | car | motor | person | rider | truck | mAP |
|---|---|---|---|---|---|---|---|---|
| Faster-RCNN Ren et al. (2015) | 34.7 | 32.0 | 56.6 | 13.6 | 37.4 | 27.6 | 38.6 | 34.4 |
| IterNorm Huang et al. (2019) | 38.5 | 23.5 | 38.9 | 15.8 | 26.6 | 25.9 | 38.1 | 29.6 |
| SW Pan et al. (2019) | 38.7 | 29.2 | 49.8 | 16.6 | 31.5 | 28.0 | 40.2 | 33.4 |
| IBN-Net Pan et al. (2018) | 37.8 | 27.3 | 49.6 | 15.1 | 29.2 | 27.1 | 38.9 | 32.1 |
| ISW Choi et al. (2021) | 38.5 | 28.5 | 49.6 | 15.4 | 31.9 | 27.5 | 41.3 | 33.2 |
| S-DGOD Wu & Deng (2022) | 40.6 | 35.1 | 50.7 | 19.7 | 34.7 | 32.1 | 43.4 | 36.6 |
| CLIP-Gap Vidit et al. (2023) | 37.7 | 34.3 | 58.0 | 19.2 | 37.6 | 28.5 | 42.9 | 36.9 |
| STAR (Ours) | **42.1** | **36.1** | **60.0** | **23.3** | **40.9** | 30.4 | 43.8 | **39.5** |

## A TRAINING ALGORITHM

The full training procedure of our proposed STAR approach is detailed in Algorithm 1.

## B EXPERIMENTAL DETAILS

In our experiments, following the previous studies Zheng et al. (2024), for the PACS dataset, a pre-trained ResNet-18 on ImageNet was fine-tuned on the source domain with images resized to $224 \times 224$. The setup included 50 epochs, batch size of 512, and a learning rate of 0.001 adjusted according to a cosine annealing scheduler. The same ResNet-18 backbone was used for the DomainNet dataset, with the experiments set to 200 epochs, and a batch size of 512, with the learning rate also following a cosine annealing pattern. All experiments were conducted with each setup replicated five times using different random seeds to ensure statistical reliability, and results were reported as the average accuracy with standard deviations. Following standard practice in single-domain generalization for object detection, we adopt Faster R-CNN Ren et al. (2015) with a ResNet-101 backbone He et al. (2016) as our detection model. Our modification is limited to the classification head, where we replace the original classification loss with a cross-entropy loss computed on target-conditioned features produced by our STO module. In addition to this modified classification loss, we incorporate two auxiliary objectives: a vision–language distillation loss, and a feature-space Mixup loss. The detection pipeline, region proposal network, and bounding box regression components remain unchanged. We train the model in 1000 iterations. The learning optimizer is SGD with a weight decay of 0.0005, and the learning rate is 0.001. In all object detection experiments, we evaluate performance using the Mean Average Precision (mAP) metric. Specifically, following the protocol in Wu & Deng (2022),

Table 8: Per-class Average Precision (AP) results for object detection on the Diverse-Weather dataset, using "Day Clear" as the source domain and "Night Rainy" as the target domain.

| Method | bus | bike | car | motor | person | rider | truck | mAP |
|---|---|---|---|---|---|---|---|---|
| Faster-RCNN Ren et al. (2015) | 16.8 | 6.9 | 26.3 | 0.6 | 11.6 | 9.4 | 15.4 | 12.4 |
| IterNorm Huang et al. (2019) | 21.4 | 6.7 | 22.0 | 0.9 | 9.1 | 10.6 | 17.6 | 12.6 |
| SW Pan et al. (2019) | 22.3 | 7.8 | 27.6 | 0.2 | 10.3 | 10.0 | 17.7 | 13.7 |
| IBN-Net Pan et al. (2018) | 24.6 | 10.0 | 28.4 | 0.9 | 8.3 | 9.8 | 18.1 | 14.3 |
| ISW Choi et al. (2021) | 22.5 | 11.4 | 26.9 | 0.4 | 9.9 | 9.8 | 17.5 | 14.1 |
| S-DGOD Wu & Deng (2022) | 24.4 | 11.6 | 29.5 | 9.8 | 10.5 | 11.4 | 19.2 | 16.6 |
| CLIP-Gap Vidit et al. (2023) | 28.6 | 12.1 | 36.1 | 9.2 | 12.3 | 9.6 | 22.9 | 18.7 |
| Ours | **29.9** | **14.4** | **38.8** | **11.9** | **14.9** | **12.0** | **25.1** | **21.0** |

Table 9: Per-class Average Precision (AP) results for object detection on the Diverse-Weather dataset, using "Day Clear" as the source domain and "Day Foggy" as the target domain.

| Method | bus | bike | car | motor | person | rider | truck | mAP |
|---|---|---|---|---|---|---|---|---|
| Faster-RCNN Ren et al. (2015) | 28.1 | 29.7 | 49.7 | 26.3 | 33.2 | 35.5 | 21.5 | 32.0 |
| IterNorm Huang et al. (2019) | 29.7 | 21.8 | 42.4 | 24.4 | 26.0 | 33.3 | 21.6 | 28.4 |
| SW Pan et al. (2019) | 30.6 | 26.2 | 44.6 | 25.1 | 30.7 | 34.6 | 23.6 | 30.8 |
| IBN-Net Pan et al. (2018) | 29.9 | 26.1 | 44.5 | 24.4 | 26.2 | 33.5 | 22.4 | 29.6 |
| ISW Choi et al. (2021) | 29.5 | 26.4 | 49.2 | 27.9 | 30.7 | 34.8 | 24.0 | 31.8 |
| S-DGOD Wu & Deng (2022) | 32.9 | 28.0 | 48.8 | 29.8 | 32.5 | 38.2 | 24.1 | 33.5 |
| CLIP-Gap Vidit et al. (2023) | 36.1 | 34.3 | 58.0 | 33.1 | 39.0 | 43.9 | 25.1 | 38.5 |
| STAR (Ours) | **38.6** | **36.1** | **58.9** | **35.3** | **41.2** | **45.1** | **27.7** | **40.4** |

we report mAP@0.5, which considers a prediction to be a true positive if it correctly matches the ground-truth class label and achieves an Intersection over Union (IoU) score greater than 0.5 with the corresponding ground-truth bounding box. This threshold provides a balanced evaluation of both localization accuracy and semantic correctness.

## C  OBJECT DETECTION PER-CLASS COMPARISON RESULTS

To better understand the strengths and limitations of each method across different semantic categories, we provide detailed per-class Average Precision (AP) results for object detection under various target weather conditions. These results allow for a finer-grained analysis of how well models generalize across challenging domain shifts, particularly for safety-critical object classes.

Table 7 presents detection results on the Night Clear scene, a particularly challenging target domain characterized by the compounded effects of low illumination. This composite shift introduces a significant discrepancy from the source (Daytime Clear), severely degrading model performance due to both photometric and structural variations. Despite these conditions, STAR achieves a 3.0% improvement in overall mAP over the strongest competing method, with notable gains of 3.3% and 1.9% in the "person" and "rider" categories, respectively. These results underscore the robustness of STAR in scenarios where conventional models struggle, validating its effectiveness under severe domain shifts.

Table 8 presents per-class detection performance when adapting from Daytime Clear to the highly challenging Night Rainy domain, which combines severe low-light degradation with weather-induced visual noise. This scenario introduces compounded appearance shifts that substantially hinder generalization. Despite this, STAR outperforms all baselines across every evaluated object category, achieving the highest overall mAP of 21.0%, a 2.3 point improvement over the strongest prior method, CLIP-Gap. Notably, STAR delivers substantial gains in safety-critical classes such as "motor" with 2.7%, "person" with 2.6%, and "rider" with 2.4% improvement, demonstrating its ability to recover object semantics under extreme visual distortion. These results highlight STAR's robustness in high-stakes, low-visibility environments where conventional models experience pronounced failures.

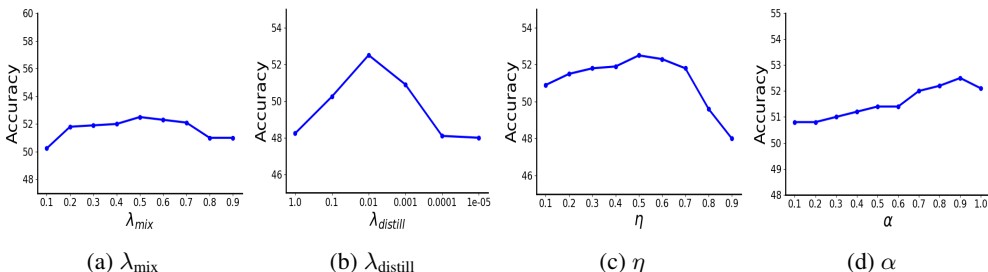

Figure 2: Sensitivity analysis for four hyper-parameters $\lambda_{\text{mix}}$, $\lambda_{\text{distill}}$, $\eta$ and $\alpha$ on PACS dataset.

Table 9 reports per-class detection performance when transferring from the Daytime Clear source domain to the Daytime Foggy target domain within the Diverse-Weather benchmark. This setting simulates real-world visibility degradation due to atmospheric scattering, which blurs contours and attenuates contrast, which are factors known to degrade detector reliability. STAR achieves the highest mean Average Precision (mAP) of 40.4%, outperforming the strongest baseline, CLIP-Gap, by 1.9 points. Across all seven object categories, STAR attains state-of-the-art results, with particularly notable gains in "motor" with 2.2%, "person" with 2.2%, and "truck" with 2.6% improvement. These improvements highlight the effectiveness of STAR's target-aware spectral conditioning in maintaining spatial and semantic precision under fog-induced ambiguity.

## D  HYPER-PARAMETER SENSITIVITY ANALYSIS

We conduct a comprehensive sensitivity analysis to evaluate STAR's robustness to key hyperparameters: the Mixup regularization weight $\lambda_{\text{mix}}$, the vision–language alignment weight $\lambda_{\text{distill}}$, the spectral truncation ratio $\eta$, and the feature fusion coefficient $\alpha$. All experiments are performed on the PACS dataset, with classification accuracy on the Cartoon domain used as the evaluation metric. As shown in Figure 2, varying $\lambda_{\text{mix}}$ from 0.1 to 0.9 reveals a peak performance at $\lambda_{\text{mix}} = 0.5$, yielding an accuracy of 52.5%. Both under-regularization and over-regularization degrade performance, with sharp drops observed beyond $\lambda_{\text{mix}} = 0.7$. This trend confirms the importance of interpolation strength calibration for effective vicinal regularization. As for $\lambda_{\text{distill}}$, Performance exhibits a sharp peak at $\lambda_{\text{distill}} = 0.01$, achieving 52.5% accuracy. Larger values (e.g., $\lambda = 1.0$) dominate the training dynamics and significantly hinder generalization, while smaller values reduce the impact of semantic supervision. This suggests that moderate alignment regularization offers the best trade-off between preserving learned representations and infusing external semantics. We observe that projecting onto the top $\eta = 0.5$ principal directions yields the best accuracy. Lower values underutilize informative variance, while higher values begin to reintroduce noise and source-specific artifacts. The symmetric drop in performance on either side of the optimum reinforces the effectiveness of low-rank semantic subspace projection. Performance improves monotonically as $\alpha$ increases, reaching its maximum at $\alpha = 0.9$, beyond which it slightly degrades. This trend indicates that high reliance on target-conditioned features is beneficial, but excessive discounting of original source features may destabilize training.

