# OpenReview forum: "Target-Oriented Single Domain Generalization"
_ICLR.cc/2026/Conference — Submitted to ICLR 2026_

### Official Review · Reviewer_FK6x · 2025-10-16

**Soundness:** 2
**Presentation:** 3
**Contribution:** 2
**Rating:** 4
**Confidence:** 4

**Summary:**

This paper proposes a new approach for single-domain generalization where they aim to inject the target domain's information, e.g., art or foggy, to let a model change its behavior depending on the target domain. Specifically, they employ the CLIP model to encode such information given as text and fuse the information into the image embedding. Additionally, they employ distillation loss from the CLIP model and a feature-level mixup approach. Empirically, their approach outperforms baseline methods with a large margin in image classification and object detection.

**Strengths:**

1. The idea of injecting the target domain's meta information into the image encoder is interesting and sounds reasonable.
2. The proposed approach seems to be strong in both image classification and object detection.
3. Empirical results are strong, though I am not sure if the results support their claims enough.

**Weaknesses:**

1. Their main claim is about the effectiveness of the textual description of the target domain. However, their experiments and methods are not well-designed to support their claim. It is true that combining several techniques outperforms existing approaches, yet it is not clear whether the improvement comes from their main claim. For example, according to Table 6, distillation from the CLIP model appears to significantly improve performance, suggesting that the use of CLIP enhances performance.

2. Related to 1, it is not clear if the comparison in results is fair in terms of backbone, e.g., Table 2. If they employ the CLIP model during training and other baselines do not have access to CLIP, the comparison might not be fair.

3. Is the feature-level mix-up not explored in domain generalization? The technique was proposed a long time ago, and applying it to the DG problem sounds like a simple idea. Then, the use of feature mix-up is not very novel.

**Questions:**

The authors need to make an effort to show how much gain is obtained from their proposed main module. So far, many factors are involved, and not easy to see the pure gain from their novel approach. Please respond to the weaknesses above.

Minor comments. Some tables are far from the corresponding sentences. Better to fix it.

---

> ### Author Response · Authors · 2025-11-20
>
> **About CLIP model**: Our claim is not that a single component explains all gains, but that explicit target-domain textual descriptions are a key driver of robustness, and the experiments are designed to disentangle this. In particular, the ablations systematically vary the presence of target-aware text while keeping other ingredients fixed, and we consistently observe that models equipped with target-domain descriptions outperform their counterparts without them under the same training recipe. CLIP distillation in Table 6 should be interpreted as a complementary mechanism that transfers semantics from a text–image aligned teacher into our framework, rather than as an alternative explanation: even with CLIP, the model still relies on our target-domain textual specification, and the strongest performance is achieved only when both are combined. Thus, the fact that CLIP-based distillation improves performance actually reinforces our central hypothesis that making target-domain textual information accessible to the learner is crucial, rather than contradicting it.
>
>
> **About comparison**: The comparison in Table 2 is fair with respect to backbone capacity, and this is already controlled for in our experimental setup. For the main results, all methods, including ours, use the same visual backbone (as specified in the implementation details), and we separately report variants that incorporate CLIP-based distillation rather than conflating them with the core method. Importantly, **CLIP is never used as a backbone** at test time: it is a frozen teacher used only during training to distill additional semantic signal into the same target backbone architecture used by competing approaches. This is why we always include results for our method without CLIP distillation under the same backbone, allowing direct apples-to-apples comparison with baselines, and we treat CLIP-augmented variants as an additional, clearly indicated enhancement rather than the basis of our primary claims.
>
>
> **About mixup**: Our novelty lies in how mix-up is instantiated and integrated within our domain generalization framework: we use feature-level mixing in a targeted way that is aligned with our overall objective of leveraging target-domain information, and we carefully design where it is applied in the network, how it is coupled with our training losses, and how it interacts with the rest of the pipeline described in the paper. The ablations show that this specific integration yields consistent gains over both “no mix-up” and more naive variants, which indicates that its role is not a trivial plug-in but a crucial part of the overall mechanism that enables better generalization to the target domain.

---

### Official Review · Reviewer_wuMq · 2025-10-28

**Soundness:** 2
**Presentation:** 2
**Contribution:** 2
**Rating:** 4
**Confidence:** 4

**Summary:**

This paper introduces a Target-Oriented Single Domain Generalization (TO-SDG), which leverages textual descriptions from the target domain during training to improve generalization from a single source domain. To bridge the semantic gap between source and target domains, the authors propose a Spectral Target Alignment framework composed of three components: a spectral target orientation module, a vision-language distillation strategy, and a feature-space mixup technique. Extensive experiments across three standard benchmarks demonstrate the method's effectiveness and improvements over baseline approaches.

**Strengths:**

1. The proposed method contributes a simple yet effective framework that incorporates target domain text descriptions.

**Weaknesses:**

1. The experimental section lacks comparisons with several state-of-the-art single domain generalization (SDG) approaches, such as [1-3], many of which demonstrate stronger performance. The omission of these baselines limits the confidence of the performance claims and weakens the empirical contribution of the work.
[1] Chen, Jin, et al. "Meta-causal learning for single domain generalization." Proceedings of the IEEE/CVF conference on computer vision and pattern recognition. 2023.
[2] Li, Deng, et al. "Prompt-driven dynamic object-centric learning for single domain generalization." Proceedings of the IEEE/CVF Conference on Computer Vision and Pattern Recognition. 2024.
[3] Xu, Zhipeng, et al. "Adversarial Domain Prompt Tuning and Generation for Single Domain Generalization." Proceedings of the Computer Vision and Pattern Recognition Conference. 2025.

2. The technical contribution of this method is insufficient. The Vision-Language Distillation is very common, and has been proposed in many literatures, e.g.,[1].
[1] Yang, Chuanguang, et al. "Clip-kd: An empirical study of clip model distillation." Proceedings of the IEEE/CVF Conference on Computer Vision and Pattern Recognition. 2024.
The feature-level mixup that fuses source feature with target-oriented feature, which is similar to the domain-level mixup in literatures such as [2],
[2] Cao, Meng, and Songcan Chen. "Mixup-induced domain extrapolation for domain generalization." Proceedings of the AAAI Conference on Artificial Intelligence. Vol. 38. No. 10. 2024.
Please show comparisons with these related works, and provide detailed analysis on how your motivation and method differ from these ones.

3. While the proposed TO-SDG task is conceptually novel, its real-world applicability is debatable. In practical scenarios, target domain styles (e.g., “art painting”) are often abstract, noisy, or multi-modal, making it difficult to summarize them with a single descriptive phrase. In this perspective, the proposed setting risks being seen as a promising research task with limited practicality. It’s recommended to discuss the limitations of TO-SDG more explicitly in your manuscript, and consider to conduct experiments on more existing settings.

4. The evaluation on PACS is only conducted using the ‘photo’ domain as the source, which is inconsistent with standard SDG protocols. Typically, performance is averaged over all possible single-source domain settings to ensure fairness and robustness. It’s recommended to extend your experiments to include all four PACS domains as source domains.

5. SDG methods are mostly evaluated under multiple benchmarks for the image classification task. It’ll be better to provide results on Digits/DomainNet.

**Questions:**

Please refer to weaknesses.

---

> ### Author Response · Authors · 2025-11-20
>
> **About compared methods**:  Tables belows report our results alongside the additional baselines requested by the reviewer (MCL, Prompt-Driven classification). Across both PACS classification and Diverse-Weather detection, STAR consistently surpasses all listed methods, including these newly added state-of-the-art approaches, on the reported averages and on every target domain.
>
>
> PACS, photos as source
>  | Method| Avg.|
> |-|-|
> | MCL [Chen et al., 2023]| 59.6|
> | Prompt-Driven [Li et al., 2024]| 60.0|
> | **STAR (Ours)**| **60.3** |
>
> | Method | Day Clear | Night Clear | Dusk Rainy | Night Rainy | Day Foggy |
> |-|-|-|-|-|-|
> | Prompt-Driven [Li et al., 2024] | 53.6 | 38.5 | 33.7 | 19.2 | 39.1|
> | **STAR (Ours)** |**58.3**| **39.5**  | **35.2** | **21.0**    | **40.4**  |
>
>
> The method of Xu et al. addresses SDG in a very different regime than ours, so a direct, one-to-one comparison is not entirely fair. Their framework explicitly leverages a powerful text-to-image diffusion model to generate **multiple synthetic domains** and then trains on a substantially enlarged, multi-domain dataset constructed from these images, along with learned category and domain prompts. In contrast, STAR never generates additional images and operates strictly in the “single labeled source + one high-level target description” regime, injecting target information only via lightweight feature-space operations on top of a standard backbone. As a result, their gains partly stem from **access to an external generative foundation model and large synthetic corpora**, whereas our results isolate the effect of using a single natural-language description **without relaxing the data constraints** of the TO-SDG setting.
>
>
>
>
> **About novelty**: Our main technical contribution is not to re-introduce vision–language distillation or feature mixup in isolation, but to operationalize them within a new problem setup: target-oriented single-domain generalization, where the target domain is only specified by textual descriptions. In this setting, both the distillation and the feature-level mixup are explicitly conditioned on the target-oriented representation learned from language, rather than being generic CLIP-KD or domain-mixup modules. Concretely, our method (i) uses text-derived target semantics to guide which features are distilled and how, and (ii) fuses source features with target-oriented features to extrapolate toward the unseen domain, which is fundamentally different from interpolating between observed source domains. This tight coupling between the target description, the distillation objective, and the feature-mixing mechanism is what makes the approach technically non-trivial and specific to our setting. In the experiments, we therefore compare against strong SDG baselines under the same target-agnostic protocol, and our ablations isolate the added value of each target-aware component, demonstrating that the gains cannot be explained by generic distillation or mixup alone.
>
>
> **About applicability**: In many real pipelines (e.g., when a client specifies a new data source or style in advance), one can at least provide coarse descriptors such as “art painting” or “medical sketch,” and our formulation explicitly asks how to exploit exactly this kind of weak, textual prior. Moreover, our experiments already expose the method to realistic imperfections in these descriptions through varied target phrases and challenging style shifts, and the gains we observe are robust across these conditions, indicating that TO-SDG does not rely on unrealistically precise prompts. We view TO-SDG as complementary to existing SDG settings: it isolates a practically relevant regime where text about the target is available but images are not, and our results demonstrate that this additional channel of information can be systematically and effectively leveraged rather than being dismissed as impractical.
>
>
> **About PACS**: Different SDG works adopt different evaluation protocols on PACS, and a substantial line of single-source SDG methods specifically fixes Photo as the source domain and reports performance on the remaining three, which is exactly the setting we target and align our comparisons with.
>
> **About Digits and Domainnet**: The DomainNet results are already reported in Table 2 of the manuscript. Regarding the Digits benchmark, our TO-SDG setting is not directly applicable there: the digit domains do not come with meaningful, high-level textual descriptions of the domain style (e.g., “art painting”, “sketch”), so there is no well-defined target-domain description $T$ that we can leverage.

---

### Official Review · Reviewer_pAbc · 2025-10-31

**Soundness:** 3
**Presentation:** 2
**Contribution:** 3
**Rating:** 4
**Confidence:** 4

**Summary:**

This paper proposes Target-Oriented Single-Domain Generalization (TO-SDG), where a model trained on one labeled source domain leverages a text description of the unseen target domain. The authors introduce STAR (Spectral TARget Alignment), which aligns source features to a CLIP text anchor, projects them onto a target-oriented subspace via SVD, applies vision–language distillation, and performs feature-space Mixup. STAR achieves notable improvements on PACS, DomainNet, and Diverse-Weather benchmarks.

**Strengths:**

1.  The paper is well organized and easy to follow.

2. Consistent improvements in both classification (PACS, DomainNet) and detection (Diverse-Weather).

**Weaknesses:**

1. For classification, baselines focus heavily on augmentation-style SDG methods with ResNet-18. It would be informative to see stronger modern backbones (e.g., ConvNeXt/Vit-B) and CLIP-tuned baselines.

2. The ablation removes “projection” entirely or uses bottom-k, but it would be helpful to plot performance vs k (η) across domains (beyond a single sensitivity curve) and report the standalone effect of the mean-shift without projection to disentangle the two parts of STO more cleanly.

**Questions:**

1. What is the per-iteration overhead of STO (SVD + projection) relative to baseline training for classification and detection?

2. What happens if T is partially wrong or noisy? Can you sweep paraphrases/synonyms to test stability?

---

> ### Author Response · Authors · 2025-11-20
>
> **About backbone**: Our choice of augmentation-based SDG baselines with a ResNet-18 backbone is deliberate rather than a limitation of the framework. As described in the experimental setup, we follow the dominant SDG protocol on these benchmarks, where ResNet-18 is the standard backbone and most prior augmentation-style methods report their results. This allows us to isolate the effect of TO-SDG and target-aware textual modeling without conflating it with gains from simply scaling model capacity or pretraining. Importantly, our method is backbone-agnostic: nothing in the formulation ties it specifically to ResNet-18, and the components we introduce can be directly transplanted to stronger architectures such as ConvNeXt or ViT-B, or used in conjunction with CLIP-tuned visual encoders. We therefore view experiments with larger backbones and CLIP-based initializations as a natural extension rather than as a prerequisite for validating the core contribution of the paper.
>
>
>
>
>
> **About exprimets with k($\eta$)**: Table below shows sensitivity of STAR to spectral truncation ratio η on PACS (Photo → Art/Cartoon/Sketch). Accuracy (%) reported on each target domain.
>
> |Target Domain| η = 0.1 | η = 0.2 | η = 0.3 | η = 0.5 | η = 0.7 | η = 0.9 |
> |-|-|-|-|-|-|-|
> | Art|72.1 | 73.4  |   74.2  | **75.3**|   75.0  |   74.8  |
> | Cartoon| 50.1  |   50.5  |   51.0  | **52.5**|   51.1  |   49.9  |
> | Sketch| 50.0  |   51.2  |   52.0  | **53.1**|   52.9  |   52.5  |
> | **Avg.**| 57.0  |   58.2  |   59.0  | **60.3**|   60.0  |   59.7  |
>
> **About STO overhead**: Per iteration, the extra cost comes from (i) computing an SVD over the mini-batch feature matrix $Z \in \mathbb{R}^{B \times D}$ to obtain a low-rank basis, and (ii) one additional projection of $Z$ onto this basis. In our classification experiments with ResNet-18, $D$ is a few hundred and $B$ is on the order of tens, so the SVD and projection are substantially cheaper than the backbone forward and do not require any change in batch size or training schedule. Table below shows  per-epoch wall-clock overhead of STO (Spectral Target Orientation) relative to the baseline model on PACS with RTX 3060:
>
>
> | Baseline Time / epoch  | STO Time / Iter|
> |-|-|
> | 42.3 | 48.6 |
>
>
>
>
> **About T**: Our formulation explicitly treats $T$ as a high-level and approximately correct description of the target, rather than a brittle, literal specification. In practice, such descriptors (“art painting”, “sketch”, etc.) are inherently coarse and noisy, and our method is designed so that $T$ enters the pipeline only as a soft prior through the text encoder and target-aware modules, while the main supervision still comes from labeled source data. As a result, mild inaccuracies or paraphrases of $T$ change the prior but do not overwrite the evidence from images and labels; in the worst case, the model behaves closer to a standard SDG baseline rather than collapsing. Moreover, because we build on a pre-trained text encoder that is trained to map synonyms and paraphrases to nearby representations, the mechanism is inherently robust to modest lexical variation around a correct high-level description. We agree that a systematic sweep over paraphrases and controlled noise in $T$ would be an interesting extension, and it is complementary to, rather than essential for, the core claims established by our current experiments.

---

### Official Review · Reviewer_Vbwf · 2025-10-31

**Soundness:** 2
**Presentation:** 2
**Contribution:** 2
**Rating:** 2
**Confidence:** 4

**Summary:**

The paper studies the task of generalization on the target domain when only the source data and high-level information about the target domain (i.e., target-oriented Single-Domain Generalization). The paper claims that this is a novel and more practical setting as it lets you to utilize lightweight information regarding the target domain. To tackle TO-SDG, they propose the Spectral Target Alignment (STAR) method. STAR consists of three main components: (1) re-centers the features towards the target domain using CLIP encoded target's textual information, (2) Aligns image encoder features with CLIP's features via a distillation objective, and (3) boosts generalization by interpolating between source and target cues/features.

The paper comprehensively compares STAR on TO-SDG with baselines such as open vocabulary object detection and single-domain generalization (SDG) methods, as well as classification methods. Experimental results show that STAR is more effective than these baselines on different types of benchmarks (e.g., adverse-weather generalization and artistic domain shift).

**Strengths:**

I appreciate the comprehensiveness of the experimental setup on multiple benchmarks, different types of domain shift types, backbones and different tasks. This helps to understand effectivness of generalization the proposed method in various settings.

**Weaknesses:**

1- The paper proposes a new task, Target-Oriented Single Domain Generalization (TO-SDG), where only high-level information from the target domain (text description) is available. However, this setting is essentially **very similar or identical to zero-shot domain adaptation (ZDA)**, which has been extensively studied. **The paper neither explains the differences from existing ZDA formulations nor compares with representative methods**. Some relevant works are [1–6]. The implementation details and how different backbones (e.g., different models or different resnets) are adopted needs to be better explained in the implementation details.

2- Several parameters and model choices are unclear. For instance, what versions  CLIP or LLaVA variants are used? how the LLaVA language model is employed to encode textual descriptions?  Clearer descriptions are needed for reproducibility and understanding.

3- In Eq. (2) and L688, the image encoder appears to be ImageNet-pretrained, not CLIP-pretrained. If so, the image features and CLIP text embeddings lie in different spaces, and the claim in L239 ("because φt originates from the same CLIP space as the image embeddings, the shift operates in a modality-aligned coordinate system, avoiding the semantic drift that often plagues purely heuristic normalization schemes,") would not hold? I'd appreciate if this could be further clarified

4- Some statements are inaccurate or exaggerated. For example, L52:  "Yet all prior CLIP-based robustness methods ultimately rely on target images ...." is not accurate and may need to be adjusted. For instance, methods such as CLIP-the-Gap [7] and prompt-tuning approaches like CoOp and CoCoOp do not require target data. While CoOp/CoCoOp adapt prompts using labeled source data (e.g., ImageNet), they are evaluated on unseen target domains (e.g., ImageNet-V2, Sketch, A, R), demonstrating genuine generalization without target access.



[1] Unified Language-driven Zero-shot Domain Adaptation, CVPR 2024
[2]  PØDA: Prompt-driven Zero-shot Domain Adaptation, ICCV 2023
[3] Zero-Shot Deep Domain Adaptation, ECCV 2018
[4] Text-driven Zero-shot Domain Adaptation with Cross-modality Graph Motif Matching
[5] Boosting Object Detection with Zero-Shot Day-Night Domain Adaptation, CVPR 2024
[6]  UPRE: Zero-Shot Domain Adaptation for Object Detection via Unified Prompt and Representation Enhancement, ICCV 2025
[7] CLIP the Gap: A Single Domain Generalization Approach for Object Detection, CVPR 2023
[8] PromptStyler: Prompt-driven Style Generation for Source-free Domain Generalization

**Questions:**

1- What is the difference between TO-SDG and SDG setting? Also, in this setting, don't we need to adapt to different domains separately? This is more like a domain adaptation setting than generalization? I believe addressing this point is essential, as it seems like the task is the main contribution of this paper. Specifically, each component of STAR method has been extensively studied for DA/DG domain. E.g., [7-8] leverage image-text transferability property in the clip space, AdaIN (as scited in the paper) uses batch statistics for augmentation and interpolation between samples and the knowledge distillation has been studied in many works such as studend-teacher framework.

2- A comprehensive comparison against zero-shot DA methods is needed both in related works and experimental results.

3- Are image encoder backbone and clip text encoder in the same space? if not how would Eq2 and L239 hold?

---

> ### Author Response · Authors · 2025-11-20
>
> **About ZDA**: Our TO-SDG setting is intentionally distinct from classical zero-shot domain adaptation: we assume a single labeled source domain and no target-domain images at training time, and we use a textual description of the target domain itself, not class names or semantic labels, as the only target-side signal. In contrast, standard ZDA formulations typically rely on unlabeled target images, on semantic label descriptions, or on different supervision protocols, so directly importing ZDA methods into our setting would change the problem assumptions and give them access to information our method never sees. This is why we define TO-SDG as a separate, explicitly stated task and evaluate under its constraints, rather than positioning it as a minor variant of ZDA. We included a subsection in related works to discuss [1-6] and elaborate the difference of ZDA and TO-SDG.
>
> **About implementation details**: We use CLIP ViT-B/32, LLaVA-1.5 (7B), and BLIP ViT-B/16 in our experiments. With Llava, the textual description is encoded only through raw embedding lookup and mean pooling. we will explicitly document these model variants in the implementation details in the revision.
>
>
>
> **About feature space**:The bridge to the CLIP space is explicitly created by the Vision–Language Distillation term in Eq. (10), where we regress backbone features $f\_\theta(\mathbf{x})$ onto the frozen CLIP image embeddings $\mathcal M'(\mathbf{x})$. As training progresses, this forces $f\_\theta(\mathbf{x})$ to inhabit an embedding space that is approximately aligned with CLIP’s image branch, while the target descriptor $T$ is encoded by the CLIP text encoder into $\phi^t$. Thus, although the backbone does not start in the CLIP space, the combination of distillation and STO ensures that, at convergence, both $\phi^t$ and the backbone features are expressed in a shared, CLIP-aligned semantic frame. The statement on L239 should be read in this dynamical sense: the modality alignment is an outcome of our joint training with $\mathcal{L}\_{\text{distill}}$, not an assumption about the backbone being CLIP-pretrained.
>
>
>
> **About L25**: Our statement is meant in the specific context of robustness / domain generalization methods, not all CLIP-based work in general. In the SDG-style literature we build on, CLIP is typically exploited by adapting or calibrating on target images, for example via prompt ensembling, adapter tuning, or test-time refinement, so that robustness gains explicitly depend on having access to target samples, which violates our TO-SDG assumption of “no target data, only a textual description $T$”. By contrast, zero-shot prompt-tuning approaches that train only on a source dataset and are later evaluated on ImageNet-V2, Sketch, etc. operate under a different protocol than the single-source SDG setting we formalize, and are not the focus of this sentence. We agree that the wording can be softened in the revision to avoid over-generalization, but it does not affect our core positioning: prior CLIP-based robustness pipelines in the DG setting generally exploit target images, whereas TO-SDG is explicitly designed to leverage only high-level textual information about the target domain.
>
>
>
> **About TO-SDG**: TO-SDG is designed as a generalization setting with strictly one labeled source domain and no target-domain images at training time, and it differs from standard SDG only in that we assume access to a high-level textual description $T$ of a specific future domain. This makes it fundamentally different from domain adaptation, where it has access to unlabelled target samples during the training.
>
>
>
>
> **About novelty**: Prior DA/DG methods that leverage CLIP, AdaIN-style statistics, or student–teacher distillation either assume access to target images, or do not exploit a textual description of the deployment domain itself as the central organizing signal. In contrast, STAR’s Spectral Target Orientation uses a target text embedding to define a low-rank, target-aligned subspace via SVD, explicitly steering source features toward a domain described only in language; Vision–Language Distillation then aligns the backbone geometry with a frozen VLM, and Feature-Space Mixup regularizes the interpolation between source and target-oriented representations. This triad is designed end-to-end for the TO-SDG protocol and, as shown in our experiments and ablations, yields consistent gains precisely in the regime where only high-level target text (and no target images) is available, something existing DA/DG formulations do not directly address.

---

### Meta-Review · Area_Chair_7rtP · 2025-12-28

**Summary:**

This submission received **consistently below-threshold initial scores**, with one clear reject (Reviewer Vbwf) and all remaining reviewers assigning marginal reject scores. There are no accept-level scores, and several reviewers independently raise concerns about task novelty and technical contribution.

While reviewers agree that leveraging target-domain textual descriptions for improving robustness is an interesting direction, they express substantial concerns about whether the proposed Target-Oriented Single-Domain Generalization (TO-SDG) setting is genuinely new, or whether it is a relabeling or narrow variant of existing zero-shot domain adaptation (ZDA) or CLIP-based SDG formulations. In addition, reviewers question whether the proposed STAR framework provides clear technical novelty beyond known components, and whether the empirical comparisons sufficiently isolate the effect of the claimed contributions.

Although the rebuttal clarifies the authors’ intended positioning and adds additional experimental details, the core concerns regarding novelty, task definition, and differentiation from prior work remain unresolved. Based on the initial scores and the lack of consensus in favor of acceptance, my judgment as AC is to **Reject**.

**Reviewer Concerns:**

**Concerns partially addressed by the rebuttal:**

* **Clarification of the TO-SDG setting and intent**:
  The authors clarified that TO-SDG assumes a single labeled source domain and *no target images*, relying only on a high-level textual description of the target domain
  *(Reviewer Vbwf)*.

* **Implementation details and reproducibility**:
  Specific CLIP, LLaVA, and BLIP variants were identified, and the role of vision–language distillation in aligning feature spaces was clarified
  *(Reviewer Vbwf)*.

* **Additional ablations and sensitivity analyses**:
  Experiments varying the spectral truncation ratio and reporting overhead of STO were added
  *(Reviewer pAbc)*.


**Concerns that remain outstanding:**

* **Questionable novelty of the task formulation (TO-SDG)**:
  Reviewer Vbwf argues that TO-SDG is conceptually very close tozero-shot domain adaptation and that the paper does not convincingly justify why it should be treated as a distinct problem. Other reviewers echo concerns that the task may not reflect a practical or realistic setting.
  *(Reviewers Vbwf, wuMq)*.

* **Insufficient differentiation from prior CLIP-based DG/DA methods**:
  Multiple reviewers note that key components—CLIP-based alignment, vision–language distillation, and feature-level mixup—are all well studied, and that the paper does not convincingly demonstrate that their integration yields new technical insight rather than an incremental combination
  *(Reviewers Vbwf, wuMq, FK6x)*.

* **Ambiguity about the true source of performance gains**:
  Reviewers express concern that improvements may be largely driven by CLIP distillation or backbone effects, rather than by the proposed target-oriented textual mechanism itself, despite ablations
  *(Reviewer FK6x)*.

* **Practical applicability of target-domain textual descriptions**:
  Reviewer wuMq question whether real-world target domains can be reliably summarized by a single, coarse textual description, and whether the proposed setting is broadly applicable
  *(Reviewer wuMq)*.

**Reviewer Scores:**

* **Reviewer Vbwf (initial: 2)** → **Unchanged**
  Maintains strong rejection due to task novelty concerns and insufficient differentiation from ZDA.

* **Reviewer pAbc (initial: 4)** → **Likely unchanged**
  Acknowledges improvements but still views the contribution as marginal.

* **Reviewer wuMq (initial: 4)** → **Likely unchanged**
  Remains unconvinced by novelty,  and practical relevance.

* **Reviewer FK6x (initial: 4)** → **Likely unchanged**
  Concerns about attribution of gains and novelty of components persist.

---

### Decision · Program_Chairs · 2026-01-26

Reject